

# Construction of competing endogenous RNA interaction network as prognostic markers in metastatic melanoma

Zan He[1,2,*], Zijuan Xin[3,4,*], Yongfei Peng[3,4], Hua Zhao[1,2] and Xiangdong Fang[3,4]

[1] Department of Dermatology, General Hospital of People's Liberation Army, Beijing, China
[2] Medical School of Chinese People's Liberation Army, Beijing, China
[3] Beijing Institute of Genomics/China National Center for Bioinformation, Chinese Academy of Science, Beijing, China
[4] College of Life Sciences, University of Chinese Academy of Sciences, Beijing, China
* These authors contributed equally to this work.

Corresponding authors
Hua Zhao, hualuck301@163.com
Xiangdong Fang, fangxd@big.ac.cn

## ABSTRACT

Malignant melanoma (MM) is a malignant tumor originating from melanocytes, with high aggressiveness, high metastasis and extremely poor prognosis. MM accounts for 4% of skin cancers and 80% of mortality, and the median survival of patients with metastatic melanoma is only about 6 months, with a five-year survival rate of less than 10%. In recent years, the incidence of melanoma has gradually increased and has become one of the serious diseases that endanger human health. Competitive endogenous RNA (ceRNA) is the main model of the mechanism by which long chain non-coding RNAs (lncRNAs) play a regulatory role in the disease. LncRNAs can act as a "sponge", competitively attracting small RNAs (micoRNAs; miRNAs), thus interfering with miRNA function, and affect the expression of target gene messenger RNAs (mRNAs), ultimately promoting tumorigenesis and progression. Bioinformatics analysis can identify potentially prognostic and therapeutically relevant differentially expressed genes in MM, finding lncRNAs, miRNAs and mRNAs that are interconnected through the ceRNA network, providing further insight into gene regulation and prognosis of metastatic melanoma. Weighted co-expression networks were used to identify lncRNA and mRNA modules associated with the metastatic phenotype, as well as the co-expression genes contained in the modules. A total of 17 lncRNAs, six miRNAs, and 11 mRNAs were used to construct a ceRNA interaction network that plays a regulatory role in metastatic melanoma patients. The prognostic risk model was used as a sorter to classify the survival prognosis of melanoma patients. Four groups of ceRNA interaction triplets were finally obtained, which miR-3662 might has potential implication for the treatment of metaststic melanoma patients, and futher experiments confirmed the regulating relationship and phenotype of this assumption. This study provides new targets to regulate metastatic process, predict metastatic potential and indicates that the miR-3662 can be used in the treatment of melanoma.

## INTRODUCTION

Malignant melanoma is derived from melanocytes, which are typically cutaneous. It is the third most common malignant tumor of the skin, but is the most malignant. The incidence of melanoma has continued to rise over the years, with approximately 300,000 new cases and 60,000 deaths from melanoma worldwide annually (*Swick & Maize, 2012*). In 2016, there were 76,380 new cases of melanoma and 10,130 patient deaths from melanoma in the United States alone (*Reddy, Miller & Tsao, 2017*). Cutaneous malignant melanomas can be classified as malignant freckled nevus-like melanoma, superficial disseminated malignant melanoma, extremity malignant melanoma, and nodular malignant melanoma based on their clinical and pathologic manifestations. Metastatic melanoma accounts for 1% of all cancer cases, but many patients experience recurrence after surgery, resulting in a poor long-term prognosis and a 5-year survival rate of less than 10% (*McKean & Amaria, 2018*; *Wahid et al., 2018*). The outcomes from current adjuvant therapies based on interferon use are not satisfactory.

A better understanding of the molecular level of MM has brought attention to the use of molecular targeted therapy targeting specific genes and signaling pathways. For example, research has led to significant results in the treatment of melanoma (*Eroglu et al., 2015*), lymphoma (*Robert et al., 2015*), Merkel cell carcinoma (*Engels, 2019*) and other tumors from relieving the immunosuppression of CD8+T cells in the microenvironment (*Topalian et al., 2016*), immunotherapy drugs for cytotoxic T lymphocyte antigen-4 (CTLA-4), and programmed cell death-1 (PD-1). However in general, the current clinical data from immunotherapy shows that its response rate is limited, and some patients have an initial drug resistance (*Lei et al., 2020*). The occurrence and development of the tumor is due to the combined effect of multiple gene mutations and the activation of multiple signaling pathways; thus, it is difficult to inhibit tumor proliferation simply by targeting a single site. A combined targeted therapy that focuses on multiple sites and pathways should be the focus of research in the future.

LncRNAs are non-coding RNAs with a transcript sequence length of 200 bp or more. These are closely related to the cell cycle and differentiation, development, reproduction, gender regulation, aging and numerous human diseases. MiRNAs are non-coding RNAs of 18–24 nt in length that are mainly involved in the regulation of individual development, apoptosis, proliferation and differentiation through complete or incomplete pairing with the 3′UTR of the target genes to degrade them or inhibit their translation. LncRNAs can competitively bind miRNAs through an miRNA response element (MRE) to inhibit the negative regulation of mRNAs by miRNAs (*Ala et al., 2013*; *Qi et al., 2015*). lncRNA is of great interest among the ceRNAs and there is growing evidence that lncRNA is involved in a wide range of biological processes. LncRNAs act as molecular sponges, competing to attract miRNAs, leading to gene silencing and are

involved in a variety of human diseases (*Prensner & Chinnaiyan, 2011*). miRNAs regulate the expression of target genes by binding to messenger RNAs (mRNAs) on target mRNAs.

The ceRNA hypothesis has attracted extensive attention in the study of the molecular biological mechanisms of cancer development. Nevertheless, a comprehensive analysis of the lncRNA-miRNA-mRNA ceRNA regulatory network based on high-throughput sequencing and large samples are still lacking in the MM field. We sought to show the specific ceRNA regulatory mechanisms in MM patients compared to non-metastatic patients, construct a ceRNA network of melanoma marker prognostic model, and provide a new perspective for elucidating the regulatory mechanisms and prognosis of metastatic melanoma by extracting the lncRNA, miRNA, and mRNA expression matrices of skin cutaneous melanoma (SKCM) *in situ* and metastatic samples from the TCGA database, and obtaining the hub genes.

## MATERIALS & METHODS

All bioinformatics analyses were conducted using RStudio (R-4.0.0). Our analyses and study design are shown in Fig. 1A. The interaction networks of Figs. S2 and S3 were drawn using Cytoscape 3.7.

### Patient cohort information and data processing

The melanoma patient cohort data were obtained from The Cancer Genome Atlas (TCGA), which is deposited on the GDC website (https://portal.gdc.cancer.gov/repository). Three files were downloaded containing miRNA, isoform, and clinical information, respectively. A total of 470 melanoma samples were stored in the database, including 103 *in situ* melanoma samples and 367 metastatic melanoma samples. The data were downloaded using the "download" command from the recommended download tool gdc-client (https://docs.gdc.cancer.gov/Data_Transfer_Tool/Users_Guide/Data_Download_and_Upload/). The clinical information and miRNA expression matrix were obtained using the R package "XML". mRNA and lncRNA were extracted separately according to the genome annotation file. The mRNA extraction parameters were type = "gene", gene_biotype = "protein_coding", and gene_biotype = "protein_coding", the lncRNA extraction parameters arguments were type = "transcript", transcript_biotype = "lncRNA". "3prime_overlapping_ncRNA", "bidirectional_promoter_lncRNA", "sense_. overlapping", "sense_intronic" and "antisense_RNA".

### Identification of differentially expressed genes

We used the R package "edgR" to identify differential genes between metastatic and *in situ* melanoma samples (*Robinson, McCarthy & Smyth, 2010*). All q-values were corrected for statistical significance of multiple testing using FDR, with screening thresholds of | log2FC| > 1.0, FRD < 0.05, and retention of significantly different genes (*Lai, 2017*; *McCarthy, Chen & Smyth, 2012*).

### Functional enrichment analysis

ClusterProfiler was used for GO/KEGG analysis (Gene Ontology/Kyoto Encyclopedia of Genes and Genomes) (*Ashburner et al., 2000*; *Kanehisa et al., 2010*; *Yu et al., 2012*). GO

 

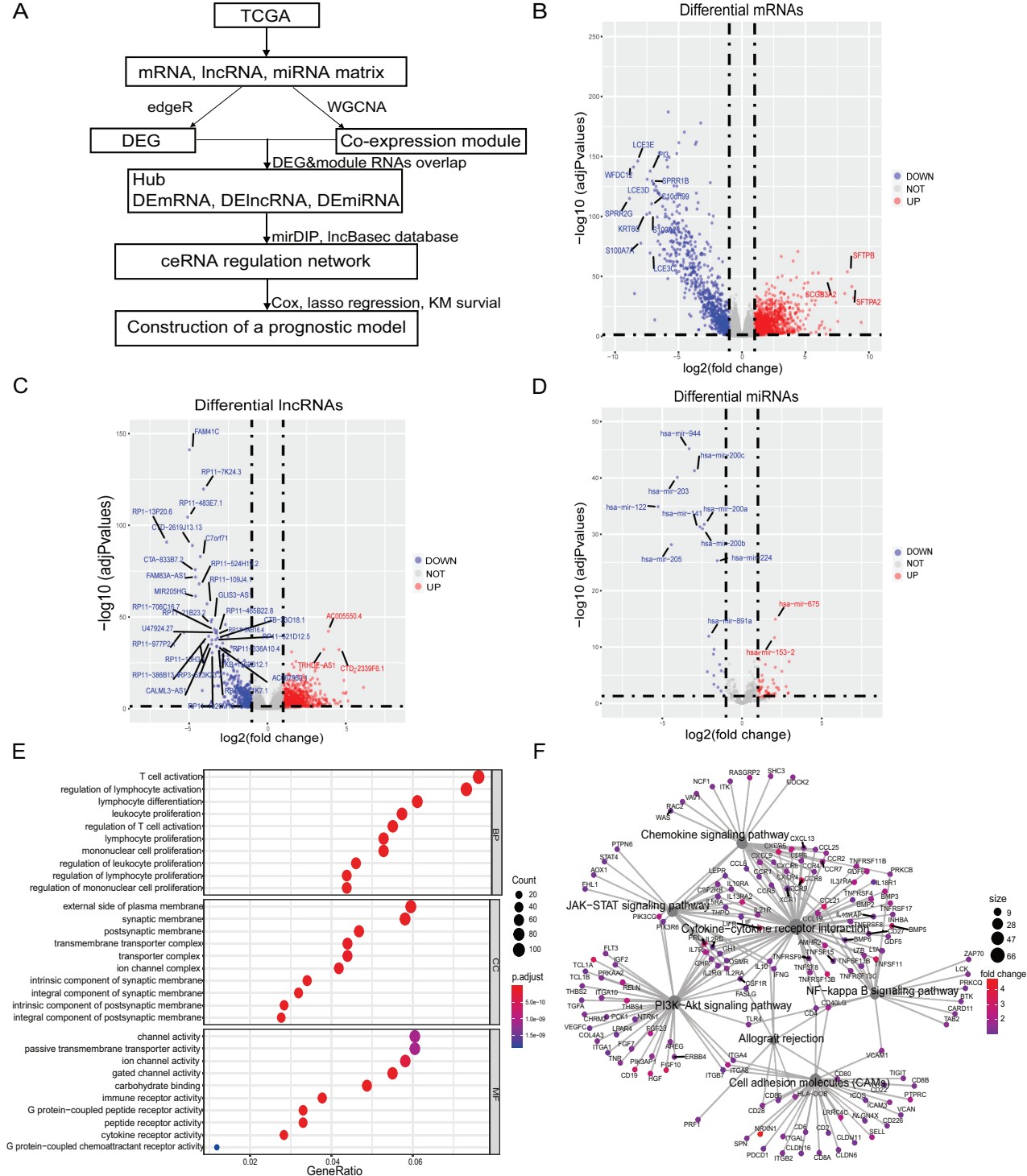

**Figure 1 Differential genes of *in situ* and metastatic samples from melanoma patients in the TCGA database.** (A) Flow chart of the bioinformatics analysis. (B) Differential mRNAs, (C) differential lncRNAs and (D) differential miRNAs. Blue represents *in situ* melanoma sample highly expressed genes, red represents metastatic samples highly expressed genes, screen |log2 fold change| > 1.0, *p*-value < 0.05. (E) Metastatic samples highly expressed DEmRNAs were subjected to GO analysis. BP, CC, and MF functionally enriched, bubble map bubble size indicates the number of genes corresponding to this entry, bubble color correlates with P. adjust, (F) Metastatic samples highly expressed DEmRNAs are enriched by KEGG to tumor-associated pathways and highly correlated genes, pathway point size represents the number of differential genes it contains, and gene point color correlates with differential multiplicity.

was used to describe gene functions along three aspects: biological process (BP), cellular component (CC), and molecular function (MF). KEGG was searched for pathways at the significance level of $p < 0.05$.

## Weighted correlation network analysis

WGCNA is an algorithm used for the identification of gene co-expression networks for mRNAs or lncRNAs with different characteristics by high-throughput spectrum of expression. Neighborhood matrix assessment of weighted co-expression relationships between all datasets was performed using Pearson's correlation analysis. We used WGCNA to analyze mRNAs and lncRNAs to obtain the mRNAs and lncRNAs with which the metastatic melanoma samples were most associated.

## Construction of a ceRNA regulatory network

Some authoritative databases have provided relationship pairs such as lncRNA-miRNA and miRNA-mRNA, which can assist in the construction of a ceRNA network centered on miRNA, based on an experimental module and prediction module. MirDIP (http://ophid. utoronto.ca/mirDIP/index.jsp) was used to predict interactions between lncRNAs and miRNAs. LncBase (http://carolina.imis.athena-innovation.gr/diana_tools/web/index.php? r=lncbasev2%2Findex) databases were used to explore target mRNAs.

## Construction of metastatic melanoma survival risk prognosis model

Metastatic melanoma samples with low expression of miRNA, high expression of lncRNA, and mRNA in the ceRNA network were used to perform three LASSO regression modeling filters. Factors that fit the normalized lambda value model were included in the subsequent analysis. A prognostic model was constructed using multiple COX regression analyses, and a comprehensive prognostic scoring system was obtained based on these factors. The risk score was calculated as follows: risk score = $\Sigma\beta i \times expRNAi$, where expRNA is the expression level of RNA and $\beta$ is the regression coefficient derived from the multivariate Cox regression model. Melanoma patients were classified into high-risk or low-risk groups based on the median risk score cutoff value based on this risk score formula and other factors we determined. Kaplan-Meier analysis was used to determine the overall survival between the groups and there was a difference in overall survival (OS) between the two groups. $P < 0.05$ was considered significant unless otherwise stated.

## Cell line and cell culture

A375 (Catalog number: CRL-1619; ATCC, Manassan, VA, USA) cells were cultured in an incubator in Dulbecco's modified Eagle's medium (DMEM, Gibco, Grand Island, NY, USA) containing 10% fetal bovine serum (AusGeneX, Molendinar, Qld, Australia) and 1% penicillin–streptomycin (P/S; Invitrogen, Carlsbad, CA, USA) with a humidity of 5% $CO_2$ at 37 °C.

## Fluorescence *in situ* hybridization (FISH)

The A375 cells were inoculated in a 12-well plate and incubated overnight in an incubator set at 37 °C with 5% CO2. The medium in the orifice plate was absorbed and washed twice

with PBS, and then one ml 4% paraformaldehyde was added to each well, and fixed at room temperature for 15 min. We followed the manufacturer's instructions for the RNA FISH kit (Catalog number: F12201/50; GenePharma, Suzhou, China). Five probes were designed and the sequences are listed in Table S1. We added 200 μl diluted DAPI working solution to each well, and stained the samples for 20 min in dark. An anti-quenching agent was added to a clean slide, and these were observed under a fluorescence microscope (630×, oil lens).

## Plasmid, siRNA and primer synthesize

Plasmids of miRNA-has-miR-3662 and miRNA-shNC, siRNA of siRP11-594N15.3 and si-NC were bought from GenePharma. The overexpressed plasmids of the miR-3666 sequence are shown in Table S2. The primers of RP11-594N15.3, ZNF831, PKIA, CSF2RB, miR-3662, GAPDH, U6 and reverse universal primer were designed by Primer Premier 5 software and were purchased from the Beijing Genomics Institute (Table S3).

## Expression vectors transfection

A375 cells were cultured in 10 cm petri dishes at a concentration of $7 \times 10^6/8$ ml the day after transfection. Cells were stably transduced with the miR-3662 and shNC expression vectors, using Lipofectamine 2000 (Invitrogen, Carlsbad, CA, USA) according to the manufacturer's protocol. Each plate was added to 12 μg plasmids and 24 μl Lipofectamine 2000. The medium was replaced after 6 h of transfection. Each plate was added to 80 μM siRNA for the transfection of siRNA and additional steps were performed as described above.

## RNA isolation and quantitative real-time PCR (qRT-PCR)

We used TRIzol® Reagent (Life Technologies, Carlsbad, CA, USA) to extract RNA from cells. After wiped off DNA, equal amounts (2 μg) of each RNA were separately reverse transcribed to their corresponding cDNAs using a RevertAid™ First Strand cDNA Synthesis Kit (Thermo Fisher Scientific, Waltham, MA, USA). All the opeations are subject to the manufacturer's instructions. CFX96 Real-Time PCR detection system (Bio-Rad, Irvine, CA, USA) was used to perform qRT-PCR with Maxima® SYBR Green/ROX qPCR Master Mix (Thermo Fisher Scientific, Waltham, MA, USA). U6 and GAPDH were used as internal control for miR-3662 and other mRNAs respectively.

## Cell counting kit-8 (CCK-8) assay

A total of 2,000 cells/well were seeded into 96-well plates, and were cultured for one, two, three, or four days. The condition of cell proliferation was assessed with a Cell Counting Kit-8 assay (C0038; Beyotime Biotechnology, China). According to manufacturer's instruction, cells were treated with 10 μL CCK-8 (100 μL culture medium per well) solution for 2 h at 37 °C after a period of incubation. Absorbance at 450 nm was measured by using a microplate reader.

## Transwell migration and invasion assays

The cells were resuspended in serum-free medium the day before experiments. For Transwell invasion assays, Transwell upper chambers (8 mm; BD Biosciences, Franklin Lakes, BJ, USA) were coated with Matrigel matrix (dilution 1:7; 356234; BD Biosciences, Franklin Lakes, BJ, USA) and $2 \times 10^5$ cells were placed in. For Transwell migration assays, $2 \times 10^4$ cells were seeded in the upper chamber of the Transwell without Matrigel matrix. Then, a medium containing 20% serum was added to lower chambers. Cells were incubated at 37 °C and 5% CO2 for 18 h, and then those cells remaining in upper chambers were removed using a wet cotton swab. We used 4% paraformaldehyde to fix the cells that had migrated and adhered to lower chambers for 10 min. Then hematoxylin (ZLI-9610, ZSGB-Bio) and eosin (ZLI-9613, ZSGB-Bio) were used to stain cells adhered to lower chambers for 20 min and then imaged. The number of cells was counted in ten separated high power fields with vertical cross distribution.

# RESULTS

## Identification of differential genes in metastatic and *in situ* melanoma samples

There were 103 samples of melanoma *in situ* and 367 samples of metastatic melanoma in the TCGA database (Tables S4, S5). We identified 1,413 differential lncRNAs, of which 1,059 were highly-expressed in metastatic samples and 354 in the *in-situ* samples. We identified 2,251 differential mRNAs, of which 1,442 were highly-expressed in the metastatic samples and 809 in the *in-situ* samples. We identified 81 differential miRNAs, of which 60 were highly expressed in metastatic samples and 21 in the *in-situ* samples (Figs. 1B–1D, Figs. S1A–S1C). The differentially expressed mRNAs in the metastatic group were enriched for GO function and KEGG pathway, immune cell activity-related terms on biological processes (BP), transmembrane transduction structures on cellular composition (CC), G protein-coupled receptor activity, passive transport and other entries on molecular function (MF) (Fig. 1E). The KEGG pathway was enriched for the PI3K-Akt signaling pathway, chemokine signaling pathway, JAK-STAT signaling pathway, cell adhesion molecules (CAMs), allograft rejection, NF-kappa-B signaling pathway, and other tumor-related pathways (Fig. 1F).

## Identification of mRNA and lncRNA co-expression modules by weighted gene co-expression network analysis

We used weighted gene co-expression network analysis (WGCNA) to analyze the gene expression patterns of mRNA and lncRNA expression matrices extracted from metastatic and *in-situ* samples. Genes with similar expression patterns were clustered into a co-expression module (*Langfelder & Horvath, 2008*). We identified nine mRNA co-expression modules and 10 lncRNA co-expression modules in the two groups, and the number of genes contained in each module is shown in Table S6. The significantly related co-expression modules were selected according to the clustering of linkage levels and the number of co-expression associations. The mRNA clustering results showed that the black module and yellow module of *in-situ* samples, and the turquoise module of
metastatic samples were identified as highly-associated co-expression modules (Figs. 2A–2C). The lncRNA clustering results showed that the pink and red module of *in-situ* samples, and the turquoise and brown modules of the metastatic samples were identified as highly correlated co-expression modules (Figs. 2D–2F). The corresponding interactions of the first 20 co-expressed genes in the above modules are shown in Figs. S2A–S2G. GO functional enrichment of co-expression genes in the turquoise module in mRNA showed that the module genes were associated with immune responses, and were also enriched for entries associated with immune negative regulation and cell adhesion (Fig. 2G). We removed the turquoise module in mRNA and the brown/turquoise module in lncRNA for the next step of the analysis.

## Construction of ceRNA regulation network

The mRNAs from the turquoise module obtained after WGCNA analysis were intersected with the highly expressed mRNAs in the metastatic samples, and 319 mRNAs were obtained. lncRNAs from the brown/turquoise module were intersected with the highly expressed mRNAs in the metastatic samples, and 287 lncRNAs were obtained (Figs. 3A–3B). The above 319 mRNAs were subjected to Pearson's correlation coefficient calculation with 287 lncRNAs. The results showed that 138 mRNAs had potential interactions with 89 lncRNAs when cor > 0.5 and $p$-value < 0.05 (Fig. S3). Using the mirDIP database to predict the target genes, the low expression miRNAs in the 21 metastatic samples were predicted to score "very high" of 3,771 mRNA target genes. The 3,771 target genes were cross-linked to 138 mRNAs by correlation analysis to obtain 11 mRNAs (CD53, CD96, CSF2RB, CXCL9, MS4A1, PIM2, PKIA, PTPN22, SH2D1A, SLA, ZNF831) (Fig. 3C). The 11 mRNAs and their corresponding six miRNAs (has-miR-346, has-miR-3662, has-miR-429, has-miR-891b, has-miR-892a, has-miR-944) were analyzed and mapped for the interaction network (Fig. 3D). These six miRNAs were used to predict 3,760 related lncRNAs in the lncBase database. The 3,760 lncRNAs were intersected with 89 lnRNAs screened in correlation analysis to obtain 17 lncRNAs (AC007386.4, AC104024.1, GTSCR1, LINC00402, LINC00861, MIAT, PRKCQ-AS1, RP11-121A8.1, RP11-164H13.1, RP11-202G28.1, RP11-203B7.2, RP11-594N15.3, RP11-598F7.3, RP11-693J15.5, RP11-768B22.2, RP5-887A10.1) (Fig. 3E). Interaction networks were mapped for the above 17 lncRNAs and six miRNAs (Fig. 3F). The final selection of 11 mRNAs and lncRNAs, the data tSNE downscaling analysis, were better able to distinguish the *in-situ* samples from the metastatic samples (Fig. 3G), resulting in the construction of lnRNA-miRNA-mRNA ceRNA network with the final selection of six miRNAs (Fig. 3H). The factors of our network were found to be highly expressed in metastatic samples in another melanoma cohort (GEO database; accession number: GSE65904) (Fig. 3I).

## Construction of metastatic melanoma survival risk prognosis model

We clustered the 34-factors included in our ceRNA interaction network, and 17 lncRNAs, six miRNAs, and 11 mRNAs were subjected to risk prediction by LASSO regression, respectively. The regularization term controlled for the lncRNA, miRNA, and mRNA

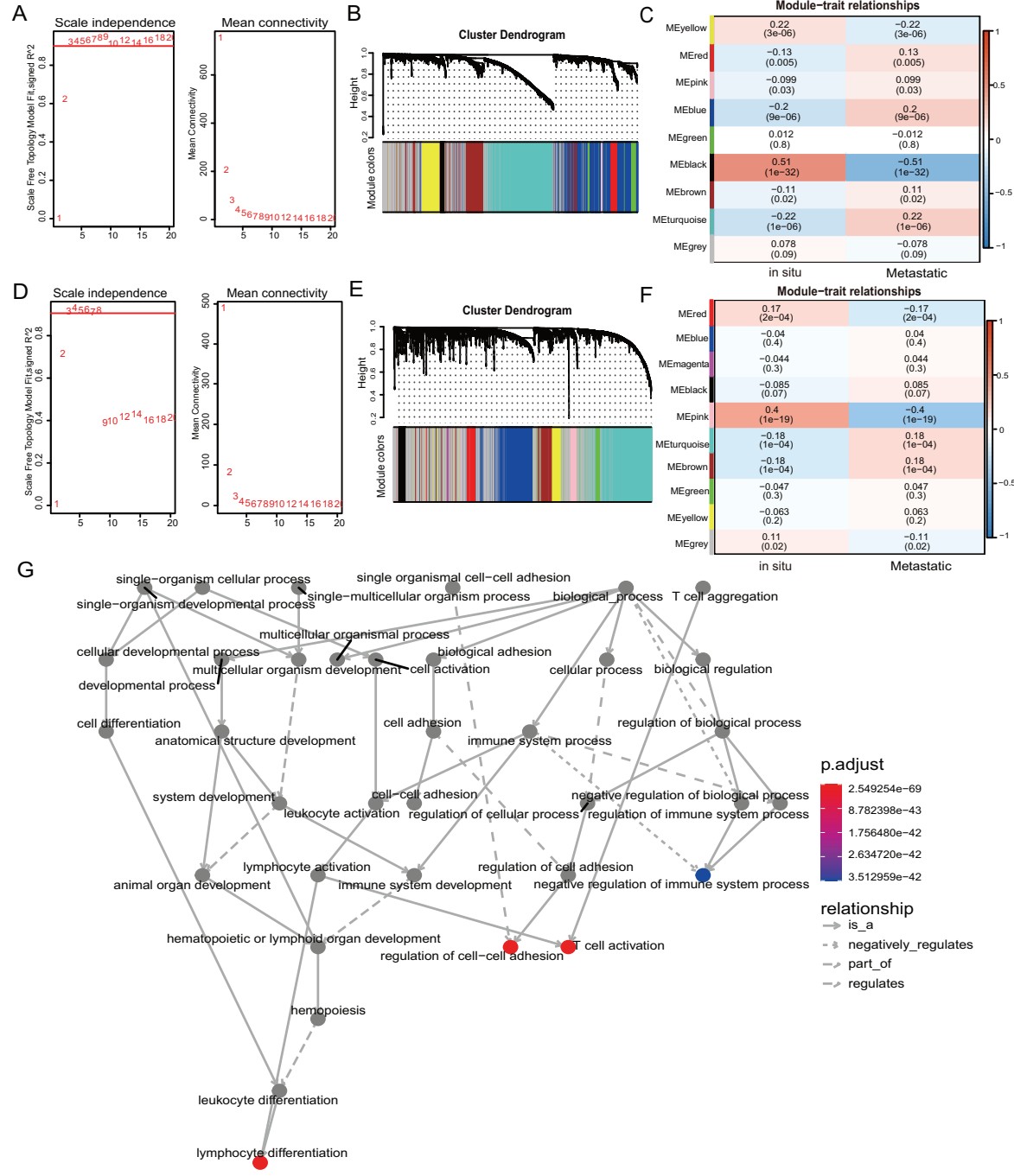

**Figure 2  WGCNA of mRNAs and lncRNAs.** (A) The scale-free fit index for soft-thresholding powers of mRNA. The soft-thresholding power in the WGCNA was determined based on a scale-free R2 (R2 = 0.95). The left panel presents the relationship between the soft-threshold and scale-free R2. The right panel presents the relationship between the soft-threshold and mean connectivity. (B) A dendrogram of the mRNA clustered based on different metrics. Each branch in the Fig. represents one gene, and every color below represents one co-expression module. (C) A heatmap showing the correlation between the mRNA module and clinical traits. The correlation coefficient in each cell represented the correlation between gene module and the clinical traits, which decreased in size from red to blue. (D) The scale-free fit index for soft-thresholding powers of lncRNA. (E) A dendrogram of the lncRNA clustered based on different metrics. (F) A heatmap showing the correlation between the lncRNA module and clinical traits. (G) The 1,255 mRNAs in turquoise module were annotated with GO function and important items were selected for drawing DAG (directed acyclic graph) graph, which shows subgraph induced by most significant GO terms.

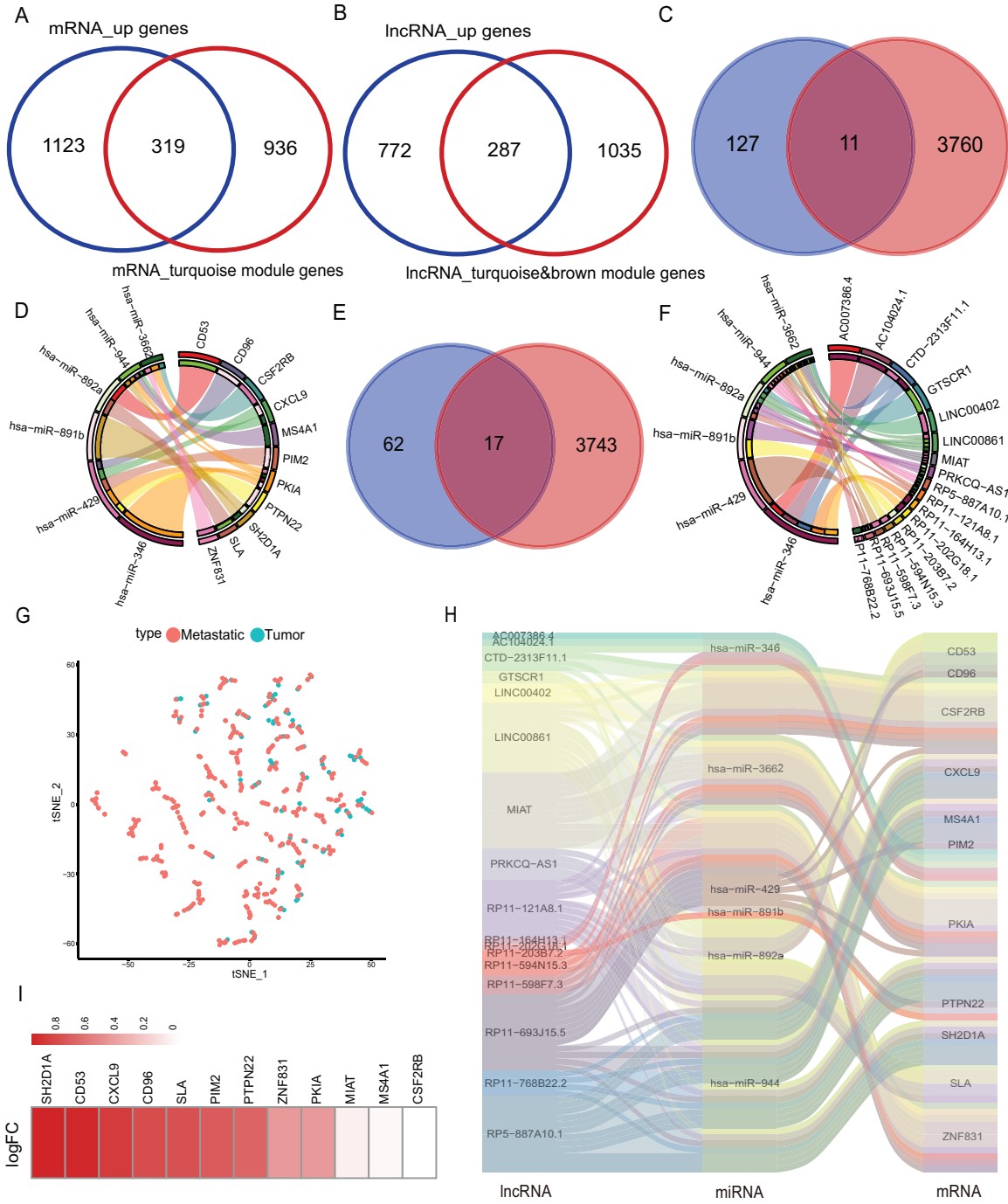

**Figure 3 Selection of hub genes and construction of the ceRNA network.** (A) The mRNAs of turquoise module in WGCNA was intersected with highly expressed mRNA from metastatic samples in the TCGA database to obtain 319 DEmRNAs, (B) the lncRNAs of turquoise module in WGCNA and brown module were intersected with highly expressed lncRNAs from metastatic samples in the TCGA database to obtain 287 DElncRNAs. (C) The mirDIP database predicted 3,771 target genes intersected with 138 highly correlated DEmRNAs, yielding 11 hub mRNAs, (D) 11 hub mRNAs interacting with their corresponding six miRNAs in the correlation network. (E) The lncBase database predicts 3,760 lncRNAs corresponding to six hub miRNAs, and 17 hub lncRNAs were obtained by intersecting with 89 DElnRNAs selected in correlation analysis, (F) 17 hub lncRNAs interacted with six hub miRNAs in a network of relationships. (G) T-SNE downscaling of TCGA data with core lncRNAs, and differences in the distribution of transfer samples and *in situ* samples. (H) Hub ceRNAs interactions network. (I) Differential expression of genes in cohort GSE65904.

models having minimal likelihood error when λ = 7, 1, and 2, respectively. We screened lncRNAs, one miRNA, and two mRNAs for high correlation with prognosis (Figs. 4A–4F). The above 10 factors were analyzed using multiple COX regression to construct risk models. There were five factors considered to be risk factors for prognosis and the other five factors were protective factors for prognosis. lncRNA AC104024.1, MIAT, RP11-598F7.3, and miRNA hsa-mir-429 could be used as significant risk markers ($P < 0.05$). The concordance index of the regression model was 0.753 (Fig. 4G). In the cohort, the expressions of AC104024.1, AC007386.4, and has-mir-429 were gradually up-regulated as the risk scoring increased, while the expressions of RP11-598F7.3, PKIA, CXCL9 and MIAT were down-regulated (Fig. 4H). The KM curves of melanoma patient survival were plotted according to the expression profiles of risk and protective factors in the multivariate COX regression model. These showed that the altered expression of factors in the model was significantly correlated with the follow-up prognosis of the cohort of patients ($P < 0.0001$) (Fig. 4I). Four groups of ceRNA interacting triplets were obtained (RP11-594N15.3–miR3662-CSF2RB, RP11-594N15.3–miR3662-ZNF831, RP11-594N15.3–miR3662-PKIA, AC104024.1–miR346-PKIA), of which miR3662 may have potential for the treatment of metastatic melanoma (Fig. 4J).

## miR-3662 down-regulates its target mRNAs and suppresses melanoma cell proliferation, migration and invasion *in vitro*

In order to verify whether the regulatory relationship of ceRNA was objective, we chose three groups of ceRNA interacting triplets (RP11-594N15.3–miR3662-CSF2RB, RP11-594N15.3–miR3662-ZNF831, RP11-594N15.3–miR3662-ZNF831, RP11-594N15.3–miR3662-PKIA) for experimental verification. The localization of lncRNA in the cytoplasm is the premise of its spongy function. The database "lncBase" was used to predict the interaction of lncRNA-miRNA based on a base experimental module and prediction module. The binding sites on the sequence between lncRNAs and miRNAs, and the interaction between lncRNAs and miRNAs was tested in other studies. Since miRNAs only exist in the cytoplasm, the lncRNAs predicted by the database must also theoretically exist in the cytoplasm. We conducted FISH experiment in melanoma A375 cells and selected lncRNA RP11-594N15.3 for localization to ensure that our lncRNAs conformed. Our results show that RP11-594N15.3 may be localized in the cytoplasm (Fig. 5A). Since RP11-594N15.3 is an RNA with a length of more than 3 KB, it is very difficult to construct its overexpression vector. Therefore, we decided to knock down the expression of RP11-594N15.3 with siRNA in A375 cells to observe the changes in the expression of miR-3662, CSF2RB, PKIA and ZNF831 (Fig. 5B). miR-3662 was found to be significantly overexpressed, PKIA and ZNF831 were down-regulated, and CSF2RB was not significant (Figs. 5C, 5D). Our experiments proved that the sponge function of lncRNA was objective.

In order to confirm whether the regulation of miR-3662 on target mRNAs in ceRNA regulatory triplets was objective, we transfected the miR-3662 overexpression plasmid vector and the NC plasmid vector into melanoma A375 cells, obtained A375-miR3662 and A375-NC cells, and verified the high expression of miR-3662 at the transcriptional level

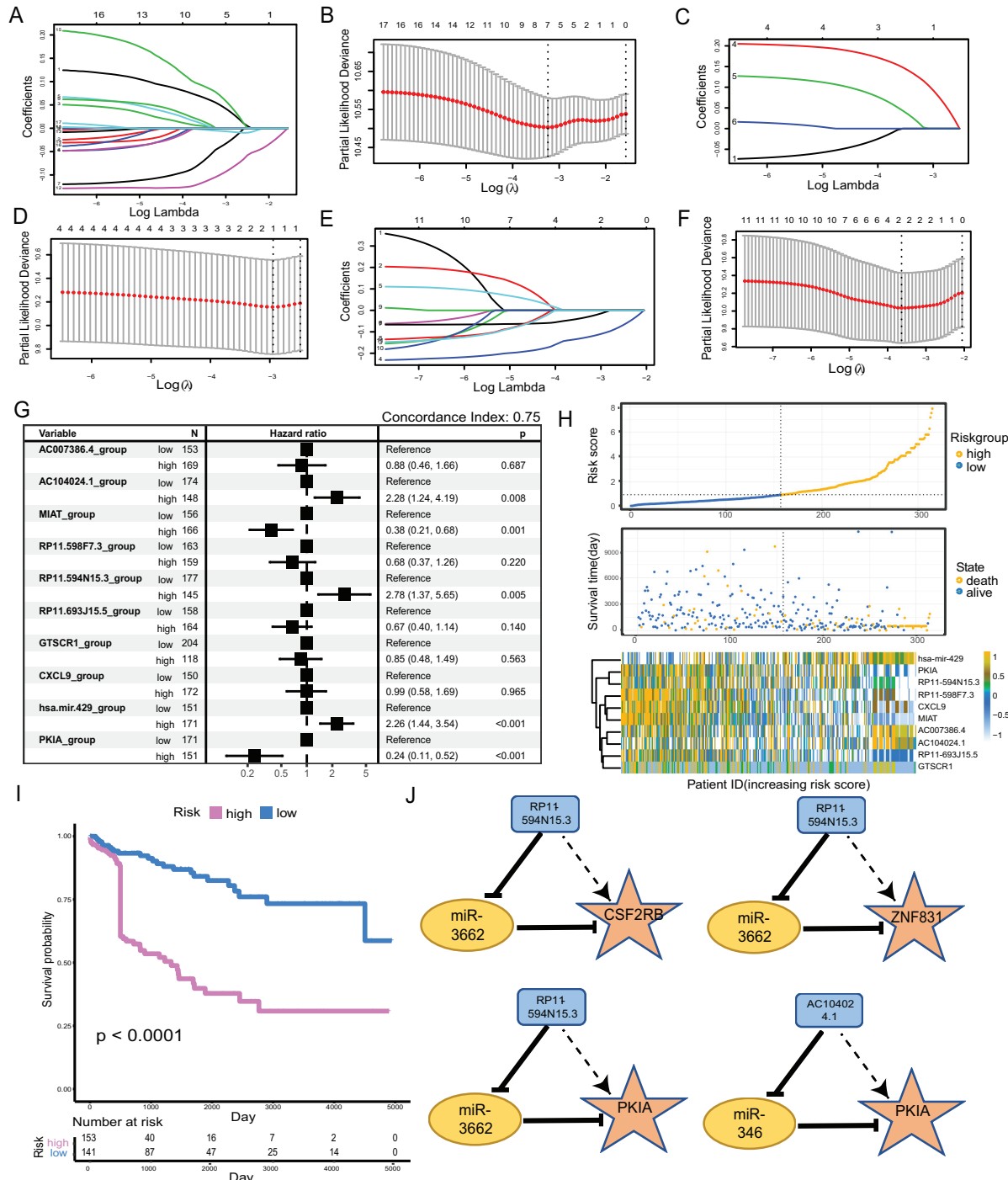

**Figure 4** **Construction of metastatic melanoma survival risk prognosis model.** (A, B) Lasso regression of 17-lncRNAs,when λ = 7,model has the best degree of fitting; (C, D) lasso regression of 6-miRNAs, when λ = 1, model has the best degree of fitting; (E, F) lasso regression of 11-mRNAs, when λ = 2, model has the best degree of fitting; (G) 10-factors were analyzed using multiple COX regression to construct risk models. Five of the factors were considered to be risk factors for prognosis and the other five factors were protective factors for prognosis. (H) The distribution of RS; the survival duration and status of patients, and a heatmap of IRGs in the classifier. (I) KM curve of the TCGA cohort according to the expression profiles of risk factors and protective factors in multivariate COX regression models. (J) The four groups of ceRNA triplets corresponding to high risk factors.

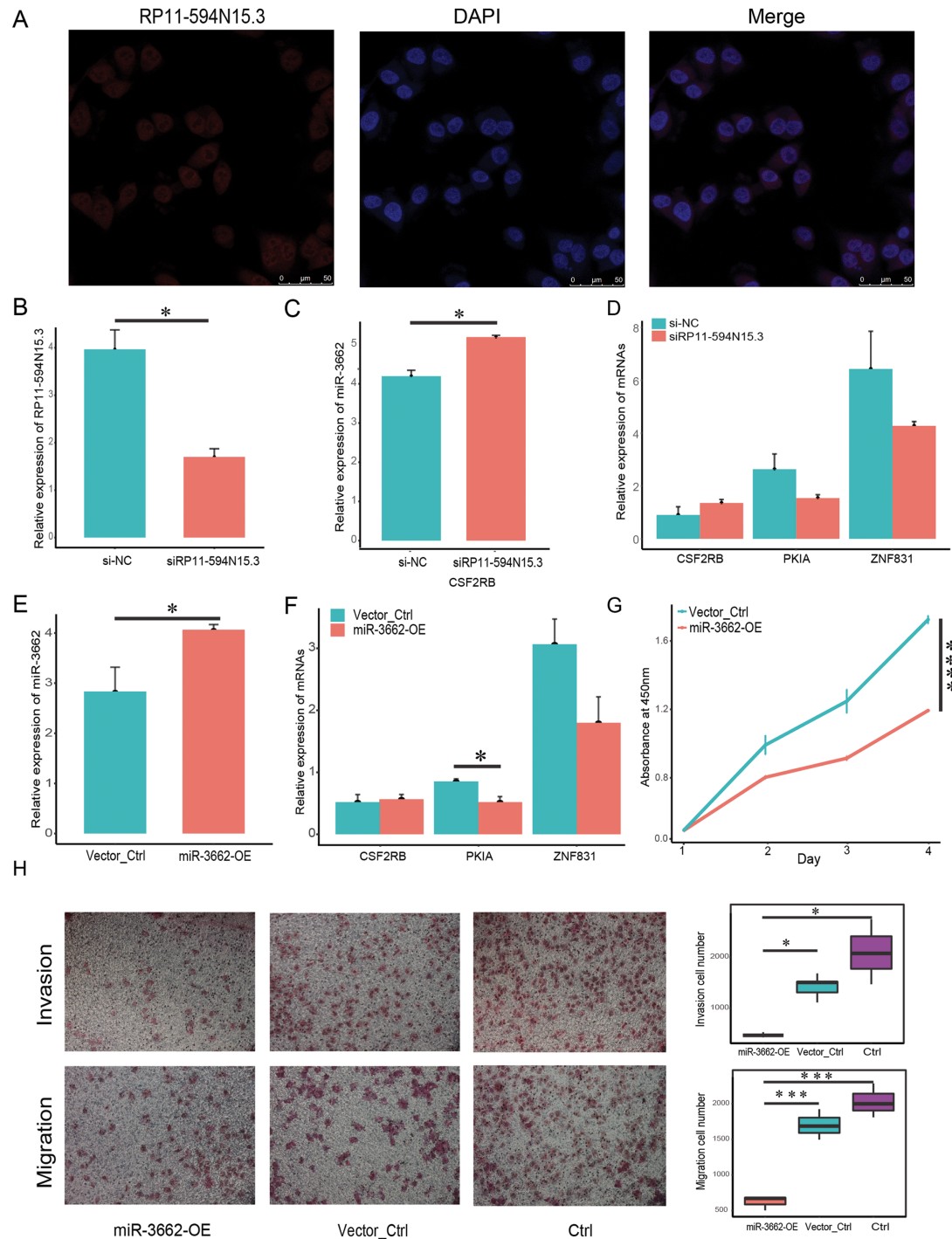

**Figure 5  miR-3662 upregulation suppresses melanoma cell migration and invasion.** (A) Localization of lncRNA RP11-594N15.3 in A375 cells by FISH assay. (B) LncRNA RP11-594N15.3 was knocked down in A375 cells, qRT-PCR analyses were used to detect its expression level. (C) QRT-PCR analyses were used to detect miR-3662 expression when RP11-594N15.3 was knocked down. (D) QRT-PCR analyses were used to detect the target mRNAs expression when RP11-594N15.3 was knocked down. (E) A375 cells were transfected with plasmids has-mir-3662 and shNC, qRT-PCR analyses were used to detect miR-3662 expression in transfected cells. (F) QRT-PCR analyses were used to detect the target mRNAs expression in transfected cells. (G) CCK-8 proliferation assay was performed on A375 miR-3662 overexpress and control cells. (H) Transwell cell migration and invasion assays were performed on A375 miR-3662 overexpress and control cells. $^*P < 0.05$, $^{***}P < 0.001$, $^{****}P < 0.001$.

(Fig. 5E). The qRT-PCR of the downstream target mRNAs shows the transcriptional differences of the three targets. The transcriptional levels of PKIA and ZNF831 were down-regulated by miR-3662, but there was no significant change on CSF2RB (Fig. 5F). The overexpression of miR-3662 significantly promoted cell proliferation (Fig. 5G). We used the Transwell chamber assay to explore the functions of miR-3662 on the invasion and migration of melanoma cells. The overexpression of miR-3662 significantly suppressed cell migration through a permeable filter and invasion through Matrigel Matrix (Fig. 5H). Our results indicate that miR-3662 may repress melanoma cell proliferation, migration and invasion by down-regulating PKIA and ZNF831.

## DISSCUSSION

The clinical treatment of melanoma poses a significant health and economic burden. There is a need to develop new chemotherapy drugs, surgical procedures and patient care (*Meng et al., 2018*). The ability of tumor cells to migrate and metastasize is an important feature in the pathogenesis of melanoma. The accurate diagnosis of early primary melanoma and timely surgical resection can significantly improve patient survival. However, the long-term survival of patients with metastatic melanoma is very low (*Filippi et al., 2016*). In this complex regulatory process, mRNAs in the tumor cell and the proteins they encode are aberrant, and the non-coding RNAs and miRNAs involved as upstream regulators also show specificity in their expression profiles, while lncRNAs, as non-coding RNAs, are involved in regulating gene and protein expression at the epigenetic level (*Wilusz, Sunwoo & Spector, 2009*). It has been phenotypically demonstrated that miRNAs can influence the progression of melanoma by inhibiting mRNA. For example, miR-331 can inhibit the proliferation and metastasis of melanoma cells by downregulating AEG-1 (*Chen et al., 2018a*).

We used The Cancer Genome Atlas (TCGA) database to obtain bioinformatics that were used to construct a differentiated endogenous competitive RNA network among patients with melanoma metastasis. We conducted a risk assessment of key factors in the network to find prognostic markers. The Weighted Gene Co-expression Network Analysis (WGCNA) was used to find gene modules for collaborative expression and to explore the relationships between gene networks and phenotypes of interest. We selected the highly expressed ceRNA in the metastatic samples after the intersection of the genes and differential genes in the co-expression module recognized by WGCNA. The lncRNA and miRNA were predicted through the mirDIP and lncBase database, and the interactive relationship between miRNA and mRNA was established to construct the overall ceRNA interaction network of DElncRNA–DEmiRNA–DEmRNA in the metastatic melanoma samples. The 17 highly expressed DElncRNAs were selected from metastatic melanoma patients and triggered the down-regulation of the expression of their corresponding 6 DEmiRNAs by exerting their sponge function. This led to the high expression of the 11 target gene DEmRNAs in metastatic patients and exerted a regulatory effect on the proliferation and invasion of melanoma cells in the process of transcription. We provided a new perspective by elucidating the upstream molecular mechanisms of melanoma invasion and migration.

We conducted three LASSO regression modeling screening of 34 factors according to lncRNA, miRNA and mRNA, respectively, in order to explore the correlation between factors in the ceRNA network and the prognosis of the cohort samples. The 10 factors were screened for best fit to the model with regularized lambda values. The risk factors of the 10 factors were modeled using multiple COX regression, and the results showed that three lncRNAs (AC104024.1, RP11-594N15.3, MIAT), one miRNA (miR-429), one mRNA (PKIA) were identified as molecular markers with high correlation with patient prognosis by constructing the risk model. AC104024.1, RP11-594N15.3 and miR-429 were risk factors for prognosis; MIAT PKIA was a protective factor for prognosis. The above factors may be used as markers to predict metastatic potential in melanoma patients and may also serve as a target to distinguish the survival prognosis of melanoma patients. Finally, we screened four groups of ceRNA regulatory triplets present in metastatic melanoma patients, in which AC104024.1 and RP11-594N15.3 served as key endogenous sponges that upregulated the expression level of target genes CSF2RB, ZNF831, and PKIA by competitive adsorbing miR-364 and miR-3662, respectively. Our assays in the A375 cell line confirmed that the phenotype of miR-3662 was consistent with the anticipation of bioinformatics analysis. The candidate genes above were studied related to cancer progression.

*Zhu et al. (2019)* expressed miR-3662 through a lentiviral vector, which could significantly reduce the ZEB1 protein level and inhibit the growth of A375 cells *in vitro* and *in vivo*. The expression of ZEB1 induced by miR-3662 decreased, the EMT expression of A375 cells was inhibited, and the relative expression of metastatic genes was reduced. Down-regulation of ZEB1 expression by a miR-3662 lentivirus vector may significantly reduce *in vitro* and *in vivo* growth of the highly invasive melanoma cell line A375. miR-3662 was considered to be a therapeutic target in adult T-cell leukemia/lymphoma (ATLL) studies. miR-3662 knockdown inhibited the proliferation of ATLL cells, and the expression of miR-3662 was related to the antiviral resistance of ATLL cells (*Yasui et al., 2018*). miR-3662 in hepatocellular carcinoma tends to be downregulated, and the high expression of miR-3662 inhibits hepatocellular carcinoma growth by inhibiting HIF-1α (*Chen et al., 2018b*). miR-429 also plays a regulatory role in the development of various tumors. For example, it promotes the proliferation of bladder cancer cells by inhibiting CDKN2B (*Yang et al., 2017*). ZEB1 expression was regulated to promote the migration, invasion and EMT of pancreatic cancer (*Shen et al., 2019*). miR-429 may also promote the proliferation of non-small cell lung cancer by targeting DLC-1 (*Xiao, Liu & Zhou, 2016*). Our study showed that in patients with metastatic melanoma, the low expression of microRNAs, miR-3662 and miR-429, led to the upregulation of the expression of target genes, CSF2RB, ZNF831and PKIA, while the transcription of differentially expressed mRNA played a specific role in the tumor phenotype.

CSF2RB (Colony Stimulating Factor 2 Receptor Subunit Beta) is involved in multiple signaling pathways, and promotes survival, proliferation, and differentiation through JAK2/STAT5, PI3K/mTOR, MEK/ERK, and other signaling pathways (*Hercus et al., 2013*). *Cimas et al. (2020)* identified CSF2RB mutations in patients with high PD-1/PD-L1 expression using the RNA-seq of basal-like breast tumors, combined with the TCGA

database, suggesting good prognosis and correlation with immune infiltration. In a public database study of lung adenocarcinoma, *Xu et al. (2020)* identified nine genes, including CSF2RB, from three independent LUAD cohorts. In colorectal cancer patients, CSF2RB was matched with 10 miRNAs that were involved in regulating apoptosis (*Slattery et al., 2018*). Normal NF-κB activation in leukemia patients positively regulated the expression of IL-3 and granulocyte/macrophage CSF2RB, thereby promoting the proliferation and survival of CML stem cells. No studies have been conducted, to our knowledge, on CSF2RB in melanoma. The protein encoded by PKIA (CAMP-Dependent Protein Kinase Inhibitor Alpha) is a member of the cAMP-dependent protein kinase (PKA) inhibitor family. *Zhang et al. (2020)* analyzed the gene transcriptome profiles of eight GEO cohorts in thyroid cancer, performed LASSO regression models, and identified seven genes, including PKIA, that were highly correlated with recurrence data in the TCGA database. In the field of cervical cancer research, PKIA is considered an important marker for stage III cervical cancer (*Banerjee & Karunagaran, 2019*). *Hoy et al. (2020)* found that the amplification of PKIA is common in prostate cancer and was associated with reduced progression-free survival. Depletion of PKIA in prostate cancer cells results in reduced migration, increased sensitivity to nest loss apoptosis, and reduced tumor growth. By altering the activity of protein kinase A, protein kinase I can act as a molecular switch that drives the gPCR-GMAP1 s-cAMP signal toward activation of α-RAP1 and MAPK, ultimately regulating tumor growth (*Hoy et al., 2020*). As a transcription factor, ZNF831 (Zinc Finger Protein 831) is thought to regulate the immune response network with phenotypes in breast cancer (*Da Silveira et al., 2017*). The co-expression network was also identified in the H-immune subtype of triple negative breast cancer (*He et al., 2018*). ZNF831 may regulate tumor immune infiltration in melanoma and plays an important role in infiltration and immune escape from melanoma.

There are some limitations to our study although we provided a complete analysis, used reasonable analytical tools, verified our results by phenotypic assays, and the results are clinically important for elucidating the regulatory mechanism and prognosis of melanoma metastasis. We predicted the interaction network of ceRNA through the database, selected RP11-594N15.3–miR-3662 and its target mRNAs for verification, however, we failed to verify all the regulatory relationships in the network objectively *in vitro*. We constructed a risk classification model, but did not verify the universality of the model in the verification set because it was difficult to find another melanoma queue containing lncRNA and miRNA sequencing data. Finally, 2-lncRNAs in the four groups of triplets lacked the evidence of previous relevant studies in tumors. Future studies should explore its function and mechanism.

## CONCLUSIONS

We used the differentially expressed genes *in situ* and metastatic samples of melanoma, as well as the phenotypic co-expression module of metastatic melanoma to construct a ceRNA interaction network that may regulate melanoma metastasis. We established a classifier model that can predict the prognosis of melanoma patients through LASSO regression and multivariate COX regression. We found that AC104024.1 and

RP11-594N15.3 can be used as markers in patients with melanoma, indicating a poor prognosis. We obtained four groups of ceRNA regulatory triplets present in metastatic melanoma patients, in which AC104024.1 and RP11-594N15.3 served as key endogenous sponges that upregulated the expression level of target genes CSF2RB, ZNF831, and PKIA by competitive adsorbing miR-346 and miR-3662, respectively. These may regulate the metastasis of melanoma and lead to a worse prognosis. Further experiments have shown that miR-3663 can reduce the transcription of its target mRNAs, and that it inhibits the proliferation, invasion and migration of melanoma cells.

### Funding

This work was supported by the National Key Research and Development Project of the Ministry of Science and Technology of the People's Republic of China (Grant No. 2020YFC2003405), the Strategic Priority Research Program of the Chinese Academy of Sciences (Grant No. XDB38020100), the National Key Research and Development Program of China (2018YFC0910702), and the National Natural Science Foundation of China (Grant No. 81672698). The funders had no role in study design, data collection and analysis, decision to publish, or preparation of the manuscript.

### Grant Disclosures

The following grant information was disclosed by the authors:
National Key Research and Development Project of the Ministry of Science and Technology of the People's Republic of China: 2020YFC2003405.
Strategic Priority Research Program of the Chinese Academy of Sciences: XDB38020100.
National Key Research and Development Program of China: 2018YFC0910702.
National Natural Science Foundation of China: 81672698.

### Competing Interests

The authors declare that they have no competing interests.

### Author Contributions

- Zan He conceived and designed the experiments, performed the experiments, authored or reviewed drafts of the paper, and approved the final draft.
- Zijuan Xin conceived and designed the experiments, performed the experiments, analyzed the data, prepared figures and/or tables, and approved the final draft.
- Yongfei Peng analyzed the data, prepared figures and/or tables, and approved the final draft.
- Hua Zhao conceived and designed the experiments, authored or reviewed drafts of the paper, and approved the final draft.
- Xiangdong Fang conceived and designed the experiments, authored or reviewed drafts of the paper, and approved the final draft.

## Data Availability

The raw measurements are available in the Supplemental Files.

## Supplemental Information

Supplemental information for this article can be found online at http://dx.doi.org/10.7717/peerj.12143#supplemental-information.

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
