# Peer review of "Construction of competing endogenous RNA interaction network as prognostic markers in metastatic melanoma"

_PeerJ, doi:10.7717/peerj.12143_

## Round 0.1 · original submission · Major Revisions

Your study is of significant interest to us, and we strongly suggest that you consider the reviewers' comments carefully.

Reviewer 1 ·

Basic reporting

Generally well written manuscript, but some of the English language is poor and would benefit from further proof-reading. There are a number of typographical and grammatical errors in the manuscript that should be addressed during revision.

Article is suitably referenced and well structured.

Experimental design

The delineation of RNA-networks in cancer of of great interest. Here the authors analyse existing RNA-seq data from TCGA and attempt to construct ceRNA relationships between lncRNA-miRNA-mRNA and then extend these data to clinical relevance. These are bold aims and the approach used, while simple, is clearly described.

The authors have not investigated the relative expression of their main targets, however, it is unclear if they have investigated whether inverse expression is seen for the miRNA and mRNAs? Also, in terms of Kaplin HS, is there any advantage to including lncRNA and miRNA, are the mRNAs not the end point here and the only differential in terms of expression that contributes to risk? If not, what are the implications? Are other unidentified targets also important?

Validity of the findings

Conclusions are generally supported by the data, however, the authors need to address several key points:

1. Are the lncRNA investigated nuclear or cytoplasmic? The majority of lincRNA localise exclusively to the nucleus, limiting their potential as ceRNA. The authors do not consider this and should discuss this point.

2. Why do A375 cells transfected with miR-3662 expression construct display lower levels of miR-3662 (Fig 5A). Is this construct actually an inhibitor of miR-3662? The methods section is unclear here and this needs addressing. What happens to PKIA and ZNF831 expression in 5B when the RP11 lncRNA is also expressed? Does this rescue knockdown of the target by the miRNA? This is an essential experiment if the authors wish to test the validity of their ceRNA triplet.

Reviewer 2 ·

Basic reporting

Some more recent references on melanoma treatment should be cited (see comments to the authors);
English would benefit a revision throughout the text

Experimental design

The study was logically thought and state of the art methods were performed. Howver it lacks a more thorough assessment in the melanoma cell model to validate the lncRNA action as ceRNA before the network can be considered of real value.

Validity of the findings

As specified above, the functional part in the cell line model needs to be more thoroughly developed

Additional comments

The authors performed an in silico analysis to build ceRNA interaction networks that could have prognostic value for metastatic melanoma.
The rational is valid and the approach in silico is powerful counting on large published data sets and state of the art tools. However, without further experiments it is quite hard to verify whether the identified network have indeed functional relevance.
The authors should at least test the protein levels of PKIA and ZFN831 upon miR-3662 modulation in the selected melanoma cell line and ideally also testing whether the selected lncRNAs indeed acts as ceRNAs in the cellular model.
The manuscript would also greatly benefit a round of English scientific revision.

Listed below are a few minor comments.

In the intro the authors should update to the most recent available references and mention the role of immunotherapy which has been paradigm changing for metastatic melanoma.

Abstract: Change ‘affects’ to ‘affect
Abstract: Change to ‘The prognostic risk model was used..’
Abstract: in the following sentence explain better how mir-3662 is linked to the ceRNA network: ‘Four groups of ceRNA interaction triplets were finally obtained, which miR-3662’…
Abstract: please clarify the following sentence which is ambiguous: miR-3662 acts as a suppressor in the treatment of melanoma.

Intro, lines 2-4: please edit the sentence (in the skin’ is repeated three times)
Intro lines 22-26 please rephrase sentence on miRNA action; it is unclear at present.
Intro line 36 define SKMC at first mention
Methods line 53. Do you mean shRNA (expressed through plasmid -DNA vector) rather than shDNA?
Results lines 144-148 when describing DE RNAs please clarify in with the total number you mean both those upregulated and downregulated
Results line change to ‘by down-regulating PKIA and ZNF831’
Discussion lines 253, 282 what do you mean by metastaticred?
Figure 1. In panel A should the hub box contain DEmRNA, DEmiRNA, DElncRNA rather than ‘DEmRNA DEmRNa DEmRNA’?
In figure 1D and 3d legends: please clarify what you mean by transfer sample?

---

## Round 0.2 · accepted · Accept

We did note that you mention that a PKIA antibody may not be commercially available, This is not our experience, and so the authors may wish to pursue this in ongoing studies.

Reviewer 1 ·

Basic reporting

The authors have addressed my previous comments relating to basic reporting in a satisfactory manner.

Experimental design

The authors have addressed my previous comments relating to experimental design in a satisfactory manner.

Validity of the findings

The authors have addressed my previous comments relating to their experimental findings in a satisfactory manner.